# iFGF23:cFGF23 Ratio Is a Questionable Diagnostic Marker for Distinguishing Between Acute and Chronic Kidney Disease

**DOI:** 10.3390/ijms26167952

**Published:** 2025-08-18

**Authors:** Joanna Szczykowska-Miller, Tomasz Hryszko, Ewa Koc-Żórawska, Beata Naumnik

**Affiliations:** 11st Department of Nephrology, Transplantation and Internal Medicine with Dialysis Unit, Medical University of Bialystok, ul. Zurawia 14, 15-540 Bialystok, Poland; joanna.szczykowska-miller@umb.edu.pl; 22nd Department of Nephrology, Hypertension and Internal Medicine with Dialysis Unit, Medical University of Bialystok, ul. Sklodowskiej-Curie 24a, 15-276 Bialystok, Poland

**Keywords:** FGF23, iFGF23:cFGF23 ratio, acute kidney injury, chronic kidney disease

## Abstract

Elevated concentrations of FGF23 are commonly observed in patients with impaired kidney function. It has been hypothesized that acute kidney injury (AKI), in contrast to chronic kidney disease (CKD), may be associated with increased FGF23 cleavage, resulting in a decreased ratio of intact to C-terminal FGF23 (iFGF23:cFGF23). However, data on the diagnostic utility of this ratio in differentiating AKI from CKD remain limited. A single-center cohort study involving 173 patients admitted to the Nephrology Department with abnormal serum creatinine levels between March 2018 and July 2021 was conducted. Blood samples were collected within 24 h of admission to measure FGF23 concentrations using both intact and C-terminal ELISAs. The iFGF23:cFGF23 ratio was calculated and analyzed across diagnostic groups. Generalized estimating equations with doubly robust adjustment were used to account for the relevant clinical and biochemical covariates. In unadjusted analyses, patients with AKI had significantly higher cFGF23 concentrations (*p* = 0.021) and a lower iFGF23:cFGF23 ratio (*p* = 0.017) compared to patients with stable CKD. No significant difference in iFGF23 levels was observed. However, after multivariable adjustment for age, serum creatinine, markers of mineral metabolism (calcium, phosphate, and parathormone) and inflammation (CRP), the observed differences were no longer statistically significant (*p* > 0.5 for all), and the interaction terms revealed no consistent modifiers of the exposure effect. The ROC analysis demonstrated modest discriminatory ability of the iFGF23:cFGF23 ratio, with an AUC of 0.60. After robust adjustment for key confounders, the iFGF23:cFGF23 ratio does not serve as a reliable independent marker for differentiating AKI from CKD. These results were supported by the ROC analysis, reflecting limited clinical utility for this ratio as a standalone biomarker. Our findings suggest that the observed differences in FGF23 metabolism are primarily driven by underlying disturbances in mineral metabolism and inflammation rather than the acute or chronic nature of the kidney injury itself.

## 1. Introduction

Every year, 13.3 million people worldwide are affected by acute kidney injury (AKI) [1]. It occurs in approximately 15–20% of all patients admitted to the hospital [2], but its incidence among critically ill patients is more than 50% [3]. Even a slight increase in serum creatinine level causes a 4-fold greater in-hospital mortality compared to patients with preserved kidney function [4]. Moreover, for those who survive to hospital discharge, the prognosis remains poor. Almost 30% of patients do not recover kidney function after an episode of AKI [5]. Currently, the diagnosis of AKI is still based on a decrease in urine output, a gradual increase in serum creatinine level, or both, which makes the diagnosis a delayed event [6]. This lag in diagnosis highlights a critical gap in early recognition and intervention. Therefore, identifying novel, sensitive biomarkers of acute kidney injury for clinical use remains a significant unmet medical need.

In recent years, fibroblast growth factor 23 (FGF23) has become a topic of interest among researchers studying AKI. It is an osteocyte-derived hormone involved in maintaining calcium–phosphate balance. The kidney is the principal target for FGF23, where its primary physiological role in healthy subjects is to regulate urinary phosphate excretion. By interacting with its specific receptor and co-receptor, Klotho protein, FGF23 diminishes sodium-dependent phosphate reabsorption and decreases the concentration of 1,25(OH)_2_ vitamin D [7,8]. The primary stimulus for FGF23 secretion is a high serum phosphate concentration; therefore, progressively increasing levels of circulating FGF23 are observed along with declining renal function [9].

One of the factors contributing to FGF23 overproduction is the intracellular proteolytic cleavage of full-length molecules [10]. As phosphaturic activity is determined by intact FGF23, its biological effect, therefore, depends on the balance between the synthesis and degradation of the mature hormone [11]. Commercially available ELISAs quantify circulating FGF23 concentrations using two strategies, which differ in the recognized epitopes. Intact FGF23 assays (iFGF23) only measure full molecules because their antibodies capture epitopes located in the N-terminal and C-terminal fragments flanking the cleavage site. By contrast, C-terminal FGF23 assays (cFGF23) detect both biologically active FGF23 and its C-terminal fragments, as the capture antibodies recognize two epitopes in the C-terminal fragment [12]. It is proposed that the simultaneous use of both assays allows for the evaluation of the intensity of intracellular FGF23 degradation by calculating the iFGF23:cFGF23 ratio [13], because cleavage will reduce the level of circulating iFGF23 but will not affect the cFGF23 concentration.

It has been demonstrated that in patients with chronic kidney disease (CKD), the FGF23 cleavage process decreases as the glomerular filtration rate declines. In end-stage renal disease (ESRD), despite extremely high FGF23 concentrations, almost all particles are biologically active [9]. However, entirely different patterns have been reported in acute kidney injury. Studies have shown an early rise in FGF23 levels, particularly in cFGF23 concentrations, preceding any detectable changes in creatinine [14]. Yet, the biological relevance of this increase remains unclear due to the limitations of cFGF23 assays in distinguishing the active hormone from inactive fragments.

It is also important to acknowledge that comorbid conditions and environmental factors beyond kidney function may influence FGF23 concentrations. Chronic obstructive pulmonary disease (COPD) has been reported to elevate circulating FGF23 levels independently of renal status [15]. Similarly, iron deficiency [13,16], intravenous iron administration [17], and exposure to environmental toxins such as cadmium [18] have been associated with alterations in FGF23 metabolism. These factors could confound the observed relationships between FGF23 isoforms and kidney injury, making it crucial to consider their prevalence and impact when interpreting biomarker data in AKI and CKD populations.

It has been hypothesized that enhanced FGF23 cleavage may occur in AKI [19], leading to lower iFGF23:cFGF23 ratios. Despite growing interest, little is known about how FGF23 metabolism, especially the iFGF23:cFGF23 ratio, differs between AKI and CKD. Differentiating these two conditions at hospital admission remains clinically challenging, particularly in patients without recent baseline kidney function data. A clear distinction is essential, as it guides key early interventions, including fluid balance, medication use, and dialysis initiation. Therefore, this study aimed to investigate whether the iFGF23:cFGF23 ratio, as a marker of FGF23 intracellular degradation, could differentiate AKI from CKD in patients presenting to a nephrology unit.

## 2. Results

### 2.1. Baseline Characteristics

A total of 173 patients with impaired kidney function who were admitted to the Nephrology Department between March 2018 and July 2021 were enrolled in this study. From this cohort, 88 patients (50.9%) were classified into the AKI group, which included 64 individuals with de novo acute kidney injury (36.99% of the study population) and 24 individuals with AKI-on-CKD (13.87%). The remaining 85 individuals with stable chronic kidney disease (49.13%) formed the CKD group.

The median age of the study population was 67 years (interquartile range, IQR: 56–75). The age differences between the groups were statistically significant (*p* < 0.001), with the study group having a higher median age. Males predominated in the total cohort, accounting for 61.27% (*n* = 106) of participants, with no significant difference in the gender distribution between the two groups (*p* = 0.362). The prevalence of COPD was similar in the AKI (4.5%) and CKD (3.5%) groups, indicating no significant difference. A detailed summary of the baseline demographic and biochemical parameters is provided in Table 1.

### 2.2. Unadjusted Analysis

A comparison of the FGF23 concentrations revealed no statistically significant difference in the log-transformed iFGF23 concentrations between the AKI and stable CKD groups (*p* = 0.629; Figure 1). However, patients with AKI had significantly higher concentrations of log-transformed cFGF23 levels compared to those with CKD (median: 2.89 RU/mL vs. 2.69 RU/mL; *p* = 0.017; Figure 2). Additionally, the calculated log-transformed iFGF23:cFGF23 ratio also differed between the two studied populations (median = −0.26 vs. −0.12; *p* = 0.021; Figure 3), suggesting the increased cleavage of FGF23 under acute conditions. The raw and log-transformed concentration of both FGF23 isoforms and their calculated ratio are shown in Table 2.

### 2.3. Adjusted Analysis

To account for potential confounding factors, adjusted analyses using a doubly robust generalized estimating equation (DRGEE) model were performed.

#### 2.3.1. Estimating the Main Effect of Exposure (No Interactions)

The effect of acute kidney injury on various forms of FGF23 was assessed in comparison to CKD. After adjusting for age, serum creatinine, albumin, calcium, phosphate, parathyroid hormone, and C-reactive protein, no statistically significant associations were observed. For iFGF23, the estimated multiplicative effect was 0.960 (95% CI: 0.76–1.21; *p* = 0.73), corresponding to a 4.0% decrease in concentration. For cFGF23, the multiplicative effect was 1.089 (95% CI: 0.81–1.47; *p* = 0.58), corresponding to an 8.9% increase. Similarly, the iFGF23:cFGF23 ratio showed no significant difference between groups (multiplicative effect: 0.9648; 95% CI: 0.816–1.140; *p* = 0.674). These results remained consistent across the 173 complete observations analyzed.

#### 2.3.2. The Effect of Exposure with Interaction Terms

The estimated effect of AKI on iFGF23 levels was a 1.698-fold increase (95% CI: 0.03–100.84, *p* = 0.80), corresponding to a 69.8% rise when covariates were at their reference values. However, the extremely wide confidence interval and non-significant *p*-value indicated high statistical uncertainty, likely due to biological variability and moderate sample size. None of the tested interactions between exposure and covariates reached statistical significance. The estimated multiplicative effects per unit increase in covariates were age (0.999), serum creatinine (0.971), serum albumin (1.039), serum calcium (0.872), serum phosphate (0.927), PTH (1.002), and CRP (0.998), all with *p*-values greater than 0.39.

Similarly, the estimated effect of AKI on cFGF23 levels was a 0.322-fold change (95% CI: 0.01–18.43, *p* = 0.58), indicating a 67.8% decrease relative to CKD at the reference covariate values. Again, the wide confidence interval and lack of statistical significance reflect considerable uncertainty. Among interaction terms, none achieved statistical significance. Marginal associations were observed for serum phosphate (0.815, *p* = 0.07) and PTH (1.006, *p* = 0.06), suggesting potential but inconclusive modification of the exposure effect.

The estimated effect of AKI on the iFGF23:cFGF23 ratio was a 1.746-fold increase (95% CI: 0.199–15.305, *p* = 0.616), indicating a 74.6% increase when covariates were at reference values. The result was not statistically significant and characterized by a wide confidence interval. Among interaction terms, only serum calcium demonstrated a statistically significant moderating effect (0.3732, 95% CI: 0.1492–0.9337, *p* = 0.035), indicating a 62.7% reduction in the exposure effect per 1 mg/dL increase in calcium. Marginal effects were observed for serum phosphate (1.1155, *p* = 0.082) and PTH (0.9962, *p* = 0.053). Other covariates, including age, serum creatinine, serum albumin, and CRP, did not show statistically significant interactions.

### 2.4. Diagnostic Performance of the iFGF23:cFGF23 Ratio (ROC Analysis)

The ROC analysis (Figure 4) identified an optimal cut-off for the iFGF23:cFGF23 ratio of ≤0.38 to discriminate AKI from CKD. At this threshold, the sensitivity was 40% and the specificity was 84%. The AUC was 0.60 (95% CI: 0.49–0.72), indicating modest discriminatory ability, as AUC values between 0.5 and 0.7 typically reflect limited clinical utility. These results suggest that a lower iFGF23:cFGF23 ratio may be associated with AKI, likely reflecting enhanced FGF23 cleavage in the setting of acute kidney dysfunction. However, the low sensitivity limits its use as a standalone diagnostic marker, although the relatively high specificity may support its role in confirming CKD in ambiguous cases.

## 3. Discussion

Interest in FGF23 in the context of acute kidney injury dates back to 2010, when Leaf et al. reported a clinical case of a 45-year-old man with rhabdomyolysis-induced AKI. The FGF23 concentration, measured using a C-terminal assay, was significantly elevated (up to 619 RU/mL) during the first 7 days of hospitalization [20]. Subsequent studies conducted by the same research group demonstrated that elevated cFGF23 concentrations are not only reproducible in larger cohorts of cardiac surgery patients but are also more pronounced when measured using C-terminal rather than intact assays, suggesting enhanced FGF23 cleavage during AKI [21].

In this study, we investigated whether the iFGF23:cFGF23 ratio could differentiate AKI from stable CKD in patients presenting with impaired kidney function. Our principal finding is that while the unadjusted ratio was significantly lower in the AKI group, consistent with the hypothesis of increased FGF23 cleavage, this association was entirely attenuated after rigorous multivariable adjustment for key clinical and biochemical confounders. This robust null finding strongly suggests that the iFGF23:cFGF23 ratio is not an independent biomarker for this diagnostic purpose.

The discrepancy between our unadjusted and adjusted results is the central finding of this paper and requires careful interpretation. The unadjusted analysis supported the initial biological premise: AKI was associated with a lower iFGF23:cFGF23 ratio, implying enhanced proteolytic cleavage. However, the disappearance of this signal after statistical adjustment indicates that the initial observation was likely an artifact of confounding. The covariates included in our model—particularly serum phosphate, calcium, and PTH—are potent, well-established regulators of FGF23 synthesis and metabolism [22]. Our analysis demonstrates that these acute disturbances in the mineral and inflammatory milieu that accompany critical illness, rather than the diagnosis of “AKI” itself, are the primary drivers of the observed changes in FGF23 processing. This interpretation is directly supported by the one significant interaction we identified in our exploratory analysis: the effect of AKI on the iFGF23:cFGF23 ratio was significantly modified by the patient’s serum calcium level (*p* = 0.035). This finding provides a mechanistic clue, underscoring that the FGF23 axis is responding to the underlying metabolic state, which overwhelms any independent signal from the acute disease process.

Relatively few studies have directly compared FGF23 concentrations between AKI and CKD populations. Most existing reports focus on either AKI or CKD in isolation, without addressing the potential diagnostic value of FGF23 in distinguishing between these two conditions. Hanudel et al. reported a 6-fold increase in cFGF23 and a 1.7-fold increase in iFGF23 among pediatric cardiac surgery patients who developed AKI [23]. Elevated cFGF23 levels have also been shown to predict AKI and adverse outcomes in critically ill adults [24]. Moreover, high FGF23 concentrations in CKD patients are closely associated with an increased risk of faster progression to ESRD [25]. However, whether FGF23 measurements can reliably differentiate AKI from CKD in clinical practice remains uncertain.

The present data do not support the use of FGF23 levels or the iFGF23:cFGF23 ratio as independent diagnostic markers for distinguishing AKI from CKD. While unadjusted differences were observed, these were attenuated after accounting for covariates known to influence FGF23 metabolism. Moreover, receiver operating characteristic analysis demonstrated only modest discriminative ability of the iFGF23:cFGF23 ratio, further highlighting its limited utility as a standalone diagnostic tool. Given the complex regulation of FGF23 by inflammation, iron status, phosphate metabolism, and assay characteristics [26,27], cross-sectional measurements may be insufficient to capture the dynamic changes associated with acute kidney injury.

This study has several limitations that should be acknowledged. First, the single-center design may limit the generalizability of our findings to other patient populations or healthcare settings. Second, FGF23 concentrations were assessed at a single time point. Given that FGF23 levels can change rapidly during AKI, serial measurements may be required to capture the whole dynamic process. Third, the use of different units for iFGF23 (pg/mL) and cFGF23 (RU/mL) precluded direct numerical comparisons between the two forms. The iFGF23:cFGF23 ratio was used to address this limitation, but it does not fully resolve the problem of non-interchangeable units or potential assay-specific biases. Fourth, we did not measure all potential regulators of FGF23 cleavage, most notably iron status, which is a known modulator and represents a potential source of unmeasured confounding. Another potential confounder not accounted for in our analysis is cadmium exposure, which has been shown to elevate circulating FGF23 concentrations. Finally, while our sample size was moderate, it may have limited our power to detect more subtle interactions in the adjusted models, as suggested by the wide confidence intervals.

Despite these limitations, this study provides novel comparative data on FGF23 metabolism in AKI and CKD and is among the few to simultaneously assess both intact and C-terminal FGF23 forms in this context. Our work underscores a critical message for the field: the evaluation of novel biomarkers must involve sophisticated statistical methods that can dissect the complex interplay of confounding, clinically relevant factors present in ill patients.

Further studies are warranted to clarify the role of FGF23 and its cleavage products in the distinction between acute and chronic kidney dysfunction. Prospective, multi-center studies incorporating serial FGF23 measurements, a comprehensive panel of covariates including markers of iron status and inflammation, and longitudinal follow-up are needed to determine whether FGF23 has utility as a diagnostic or prognostic biomarker in patients with kidney injury.

## 4. Methods

### 4.1. Study Design and Participants

This study was designed as a single-center cohort study to evaluate the clinical utility of the iFGF23:cFGF23 ratio at the time of admission to the Nephrology Unit for the early discrimination of acute kidney injury from chronic kidney disease. The Local Ethics Committee approved the study protocol and all patients or their legally authorized representatives provided written informed consent. Study participants were recruited between March 2018 and July 2021.

The inclusion criteria were age > 18 years and an abnormal creatinine concentration at admission. Patients receiving maintenance dialysis or those who were pregnant were excluded from the study. Only patients with complete clinical and biochemical data, who provided informed consent, were included in the study. Information on patients screened but not enrolled was not systematically collected, which constitutes a limitation of the study. The patient selection process is summarized in Figure 5.

Based on the KDIGO guidelines, participants were stratified into AKI or CKD groups. Patients with chronic kidney disease who experienced an abrupt deterioration in kidney function fulfilling the KDIGO criteria for AKI, with an identifiable precipitating factor (e.g., sepsis, hypovolemia), were classified as AKI-on-CKD. These cases were analyzed together with de novo AKI, as both represent acute kidney injury episodes requiring similar diagnostic and therapeutic approaches.

### 4.2. Sample Collection and Laboratory Measurements

Venous blood samples for laboratory assessments were collected within 24 h of admission to the Nephrology Department. In patients requiring kidney replacement therapy, samples were obtained prior to the first dialysis session. None of the study participants received intravenous iron supplementation prior to blood sampling. Plasma was separated by centrifugation at 3500 rpm for 15 min at room temperature within 30 min of collection. Aliquots were then stored at −70 °C until further analysis.

Quantitative measurements of FGF23 were performed using commercially available enzyme-linked immunosorbent assay (ELISA) kits for both the C-terminal (Cat. No. 60–6100) and intact (Cat. No. 60–6300) forms of FGF23 (Immutopics, Inc., San Clemente, CA, USA), following the manufacturer’s instructions. The C-terminal assay detects both full-length FGF23 and its C-terminal fragments, whereas the intact assay measures only the biologically active hormone. Each patient had one plasma sample analyzed, and each measurement was performed once per sample without replication. Absorbance was read at 450 nm using a microplate reader, and the results were expressed in RU/mL for cFGF23 and pg/mL for iFGF23. All other laboratory parameters, including serum creatinine, urea, electrolytes, and complete blood counts, were performed using standard automated methods in the hospital’s central laboratory.

### 4.3. Statistical Analysis

#### 4.3.1. Descriptive and Nonparametric Analyses

The statistical analysis was performed with R programming language (version 4.3.1) [28] on a Windows 10 Pro 64-bit system (build 19045), utilizing the following packages: *report* (version 0.5.7) [29], *ggstatsplot *(version 0.12.1) [30], *gtsummary* (version 1.7.2) [31], *readxl* (version 1.4.3) [32], and *psych* (version 2.3.9) [33]. The normality of the data distribution was verified using the Shapiro–Wilk test. For continuous variables, data are reported as the median and interquartile range (IQR, 25th–75th percentiles). For categorical variables, the frequency and percentage of each category are presented. To compare measures of central tendency between two independent groups for a continuous variable with a non-normal distribution, the nonparametric Wilcoxon rank-sum test was applied. Differences in categorical variables were assessed using Pearson’s chi-square test. All comparisons are two-tailed, with *p* < 0.05 considered significant. To improve the homogeneity of variance, the results were log-transformed before the statistical analysis.

#### 4.3.2. Adjusted Analyses Using DRGEE (Doubly Robust Generalized Estimating Equation)

To estimate the causal effect of the exposure (AKI vs. CKD) on iFGF23 and cFGF23, a DRGEE model was employed. This approach was chosen for its ability to provide robust estimates in observational studies where confounding is a major concern. It combines an outcome regression model with a propensity score model to provide consistent estimates if either model is correctly specified, thereby enhancing robustness against misspecification bias.

The exposure was defined as a binary variable (X = 1 for AKI, X = 0 for CKD), with log-transformed iFGF23 and cFGF23 as continuous outcomes to address skewness. Covariates included in the model were age, serum creatinine, albumin, calcium, phosphate, parathyroid hormone (PTH), and C-reactive protein (CRP). Interaction terms between the exposure (AKI vs. CKD) and each covariate were included to assess whether the effect of AKI varied across the levels of these factors. An independent working correlation structure with robust sandwich standard errors was used. DRGEE models were fitted separately for iFGF23 and cFGF23, with and without interaction terms between the exposure and confounders. Multiplicative effects on the original scale (iFGF23 or cFGF23) were derived by exponentiating the log-scale estimates, and 95% confidence intervals were calculated accordingly. All analyses adhered to a significance threshold of *p* < 0.05.

The DRGEE model was fitted with a linear link function to reflect the continuous nature of the outcomes, assuming a linear relationship with the exposure and covariates. Robust standard errors were calculated to account for potential heteroskedasticity or misspecification of the variance structure, ensuring valid inference. No clustering or random effects were included, as the data represent a single measurement per patient and lack a hierarchical structure. The results are reported as point estimates of β_1_ (the exposure effect) with 95% confidence intervals.

#### 4.3.3. ROC Analysis

To evaluate the diagnostic performance of the iFGF23:cFGF23 ratio in distinguishing AKI from stable CKD, receiver operating characteristic (ROC) curve analysis was performed using the *cutpointr* package in R (version 4.4.1). The optimal threshold was identified by maximizing Youden’s index (sensitivity + specificity − 1). Values less than or equal to this cut point were interpreted as indicating a higher probability of the positive outcome (AKI). To enhance the robustness of the estimates, bootstrap resampling with 2000 iterations was applied. The area under the ROC curve (AUC) and its 95% confidence interval (CI) were computed using a nonparametric bootstrap approach. Sensitivity and specificity at the optimal cut point were reported. All analyses were two-sided with an alpha level of 0.05 and were performed within a secure computing environment.

## 5. Conclusions

This study investigated the potential of FGF23 isoforms and the iFGF23:cFGF23 ratio as diagnostic markers for distinguishing between acute kidney injury and chronic kidney disease. While unadjusted analyses indicated higher cFGF23 levels and a reduced iFGF23:cFGF23 ratio in AKI, these associations did not persist following multivariable adjustment for key clinical and biochemical covariates. The lack of statistically significant differences in the adjusted models suggests that the unadjusted associations may be confounded by underlying differences in kidney function, as well as mineral metabolism and systemic inflammation, which are themselves known modulators of FGF23 levels, rather than reflecting a distinct pathophysiological signature of AKI.

Given these findings, the clinical applicability of the iFGF23:cFGF23 ratio as a standalone biomarker for discriminating AKI from CKD appears to be limited. This conclusion is supported by ROC analysis, which demonstrated only the modest discriminative ability of the iFGF23:cFGF23 ratio, additionally underscoring its limited utility as an independent diagnostic marker. Further prospective and adequately powered studies are warranted to clarify the pathophysiological relevance and potential diagnostic role of FGF23 processing in acute versus chronic kidney dysfunction.

## Figures and Tables

**Figure 1 ijms-26-07952-f001:**
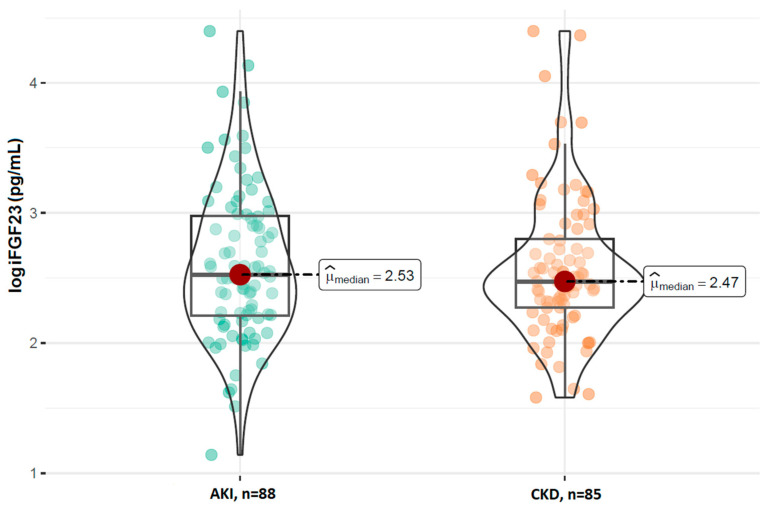
Comparison of the log iFGF23 concentrations between patients with AKI and CKD. Boxplots show median, interquartile range, and full data range for each group (AKI: n = 88, CKD: n = 85).

**Figure 2 ijms-26-07952-f002:**
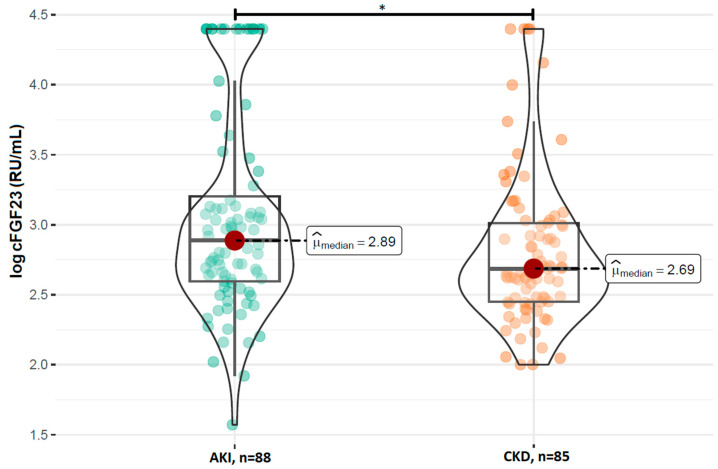
Comparison of the log cFGF23 concentrations between patients with AKI and CKD. Boxplots show median, interquartile range, and full data range for each group (AKI: n = 88, CKD: n = 85). * *p* < 0.05 indicates statistical significance between groups (Mann–Whitney U test).

**Figure 3 ijms-26-07952-f003:**
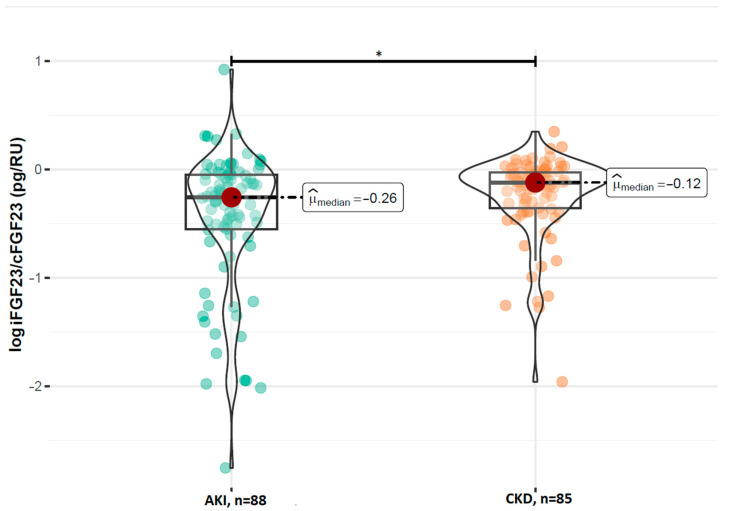
Comparison of the log iFGF23:cFGF23 ratio between patients with AKI and CKD. Boxplots show median, interquartile range, and full data range for each group (AKI: n = 88, CKD: n = 85). * *p* < 0.05 indicates statistical significance between groups (Mann–Whitney U test).

**Figure 4 ijms-26-07952-f004:**
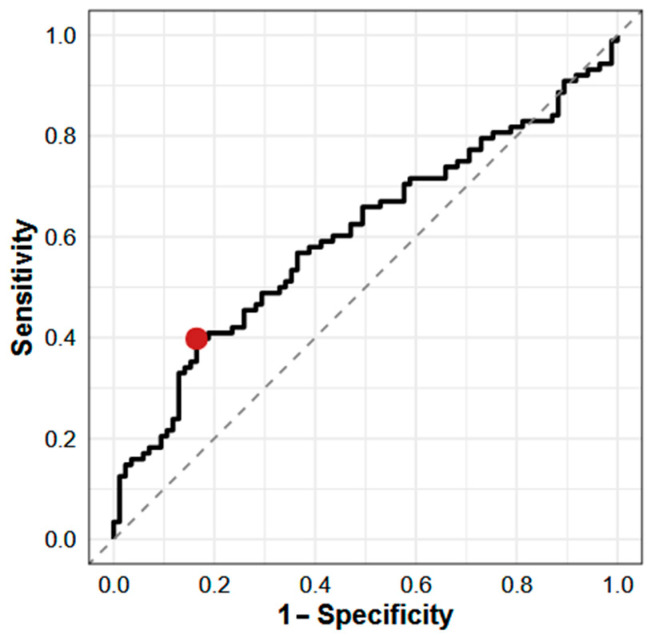
ROC curve for iFGF23:cFGF23 ratio in differentiating AKI from stable CKD. The red point indicates the estimated optimal cut point. Cut point: ≤0.38 pg/RU; sensitivity: 0.40; specificity: 0.84; AUC: 0.60 (95% CI: 0.49–0.72). AUC—area under the curve.

**Figure 5 ijms-26-07952-f005:**
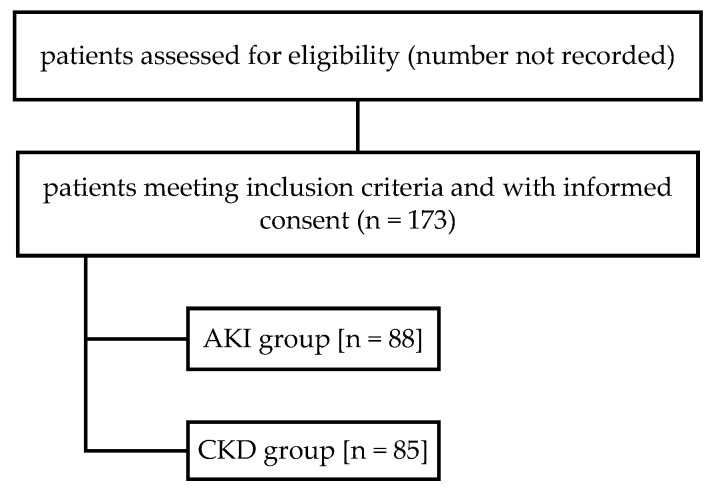
Flow diagram of patient inclusion.

**Table 1 ijms-26-07952-t001:** Baseline characteristics of the studied population. Data are presented as medians and interquartile ranges for continuous variables, or as numbers of cases and percentages. COPD, chronic obstructive pulmonary disease; CRP, C-reactive protein; s, serum.

Parameter	All (*n* = 173)	AKI (*n*_1_ = 88)	CKD (*n*_2_ = 85)	*p*
Demographic				
Age	67 (56–75)	69.5 (62.75–78)	63 (49–70)	<0.001
Female (%)	67 (38.7)	37 (42)	30 (35.3)	0.362
COPD (%)	7 (4.04)	4 (4.5)	3 (3.5)	0.734
Laboratory				
sCreatinine (mg/dL)	5.55 (3.81–7.85)	6.86 (4.44–9.05)	4.33 (3.22–6.9)	<0.001
sAlbumin (g/dL)	3.68 (3.27–3.98)	3.47 (3.02–3.8)	3.84 (3.61–4.22)	<0.001
Calcium (mmol/L)	2.17 (2.01–2.28)	2.14 (1.97–2.24)	2.20 (2.09–2.31)	0.018
Phosphate (mg/dL)	4.98 (4.22–6.37)	5.52 (4.56–6.63)	4.57 (3.85–5.79)	<0.001
Parathormone (pg/mL)	130.00 (83.14–212.20)	113.45 (77.76–142.70)	172.90 (103.2–291.40)	<0.001
CRP (mg/L)	13.29 (2.96–55.69)	39.01 (7.78–96.44)	3.80 (1.92–15.12)	<0.001
Hemoglobin (g/dL)	10.30 (9.30–11.60)	10.10 (9.30–11.33)	10.50 (9.20–11.80)	0.226

**Table 2 ijms-26-07952-t002:** Raw and log-transformed iFGF23, cFGF23, and iFGF23:cFGF23 ratios: median and interquartile range in the total cohort and subgroups. FGF2, fibroblast growth factor-23; iFGF23, FGF23 measured with intact FGF23 assay; cFGF2, FGF23 measured with c-terminal FGF23 assay.

Parameter	All (*n* = 173)	AKI (*n*_1_ = 88)	CKD (*n*_2_ = 85)	*p*
iFGF23 (pg/mL)	315.40 (164.80, 822.60)	336.00 (162.28, 948.28)	298.20 (187.20, 630.70)	0.629
log iFGF23 (pg/mL)	2.50 (2.22, 2.92)	2.53 (2.21, 2.98)	2.47 (2.27, 2.80)	
cFGF23 (RU/mL)	551.20 (313.90, 1299.00)	773.20 (392.80, 1597.00)	486.20 (282.10, 1028.00)	0.017
log cFGF23 (RU/mL)	2.74 (2.5, 3.11)	2.89 (2.59, 3.20)	2.69 (2.45, 3.01)	
iFGF23:cFGF23 (pg/RU)	0.67 (0.35, 0.93)	0.55 (0.28, 0.90)	0.76 (0.44, 0.94)	0.021
log iFGF23/cFGF23 (pg/RU)	−0.18 (−0.45, −0.03)	−0.26 (−0.55, −0.05)	−0.12 (−0.35, −0.03)	

## Data Availability

The dataset is available on request from the authors.

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
