# Peer review of "iFGF23:cFGF23 Ratio Is a Questionable Diagnostic Marker for Distinguishing Between Acute and Chronic Kidney Disease"

_ijms, 2025, doi:10.3390/ijms26167952_

Round 1
Reviewer 1 Report
Comments and Suggestions for Authors
Before the paper can be further processed, some methodological clarifications are required.
If the authors intend to compare iFGF-23, cFGF-23, and their ratio between individuals with AKI and CKD, it is essential that these two groups (AKI and CKD) do not differ in other conditions or diseases that may influence any FGF-23 isoform. At least two such factors are known: chronic obstructive pulmonary disease (COPD) [10.1016/j.rmed.2021.106404] and iron deficiency [10.1016/j.bone.2017.02.005]. Additionally, a more detailed literature search reveals that cadmium exposure may also affect FGF-23 levels [10.1093/toxsci/kfu043].
For example, if one of the groups (either AKI or CKD) has a higher prevalence of COPD, and COPD is known to elevate iFGF-23, it would not be possible to determine whether the observed FGF-23 elevation is due to COPD or CKD. Therefore, the AKI and CKD groups should not differ in at least iron status and COPD prevalence.
Please address this issue.
Author Response
Reviewer #1
Before the paper can be further processed, some methodological clarifications are required.
If the authors intend to compare iFGF-23, cFGF-23, and their ratio between individuals with AKI and CKD, it is essential that these two groups (AKI and CKD) do not differ in other conditions or diseases that may influence any FGF-23 isoform. At least two such factors are known: chronic obstructive pulmonary disease (COPD) [10.1016/j.rmed.2021.106404] and iron deficiency [10.1016/j.bone.2017.02.005]. Additionally, a more detailed literature search reveals that cadmium exposure may also affect FGF-23 levels [10.1093/toxsci/kfu043].
For example, if one of the groups (either AKI or CKD) has a higher prevalence of COPD, and COPD is known to elevate iFGF-23, it would not be possible to determine whether the observed FGF-23 elevation is due to COPD or CKD. Therefore, the AKI and CKD groups should not differ in at least iron status and COPD prevalence.
Please address this issue.
Response 1:
Thank you for this vital comment regarding potential confounders such as COPD, iron deficiency, and cadmium exposure. We agree that these factors have been reported to influence FGF23 levels.
In our cohort, the prevalence of COPD was very low and similar between groups (4.5% in AKI vs. 3.5% in CKD), making a substantial confounding effect unlikely. These data have been added to the manuscript (Results section and Table 1).
Unfortunately, iron status was not systematically assessed in our study, and data on cadmium exposure were not available. We acknowledge this as a limitation and have included an explicit statement in the Discussion section. Nevertheless, we fully agree that future studies should incorporate iron metabolism and environmental exposures into the design to better account for these potential confounders.
Changes made:
- added a paragraph about the confounding factors in the Introduction section,
- added COPD prevalence comparison in the Results and Table 1,
- added a limitation regarding iron status and cadmium exposure in the Discussion section.
Reviewer 2 Report
Comments and Suggestions for Authors
The manuscript titled “iFGF23:cFGF23 ratio as a potential diagnostic marker in acute kidney injury.” investigated the potential of the iFGF23:cFGF23 ratio as diagnostic markers for distinguishing between acute kidney injury and chronic kidney disease. However, there are many logical errors and incorrect methods. Therefore, major revisions have to be done before this manuscript could be accepted for publication in this journal.
Major issues:
- For chronic kidney disease (CKD) and acute kidney injury (AKI), there are many kinds of causes. They are classified according to different criteria, such as etiology and KDIGO guidelines. However, the author analyzes all CKD patients as a whole. I think it lacks accuracy and clinical validity. The author should group for the different etiologies of the disease.
- The goal of this study is to distinguish the difference between AKI and CKD, I feel confused about why the authors put the 64 patients with AKI and the 24 patients with acute on chronic kidney disease together in the same study group?
- In general, the performance of a diagnostic maker is evaluated using the Receiver Operating Characteristic (ROC) curve. The author should add a curve analysis.
- The author says “adjusted analyses do not support its utility as a robust diagnostic marker for distinguishing AKI from CKD” I think the conclusion diminish academic and innovative value.
Minor issues:
- Page 4, Lines 158-160: “From this cohort, 88 patients (50,9%) were classified into the study group, which included 64 individuals with acute kidney injury (36.99% of the study group) and 24 individuals with significant exacerbation of chronic kidney disease (13.87%).” The author wrote “percentage of the study group,” but it should be “percentage of the whole group.” The proportion of the 64 acute kidney injury patients with the study group should be 72.7%.
- Page 4, Line 157: “enrolled on the study” should be changed to “enrolled in the study”
- Page 4, Line 158: “50,9%” should be changed to “50.9%”.
- The sample sizes for the AKI group (n=88) and the CKD group (n=85) are too small. It can decrease statistical power.
Author Response
Reviewer#2
The manuscript titled “iFGF23:cFGF23 ratio as a potential diagnostic marker in acute kidney injury.” investigated the potential of the iFGF23:cFGF23 ratio as diagnostic markers for distinguishing between acute kidney injury and chronic kidney disease. However, there are many logical errors and incorrect methods. Therefore, major revisions have to be done before this manuscript could be accepted for publication in this journal.
Major issues:
1. For chronic kidney disease (CKD) and acute kidney injury (AKI), there are many kinds of causes. They are classified according to different criteria, such as etiology and KDIGO guidelines. However, the author analyzes all CKD patients as a whole. I think it lacks accuracy and clinical validity. The author should group for the different etiologies of the disease.
Thank you for your valuable comment regarding the classification of CKD patients by etiology. We agree that CKD is a heterogeneous condition with various causes, and that etiology can influence some disease characteristics.
However, the primary focus of our study was not on CKD as a whole, but on investigating the iFGF23:cFGF23 ratio as a biomarker to differentiate AKI from chronic kidney disease CKD. To date, the only known CKD etiology associated with distinct FGF23 levels, especially at earlier stages, is autosomal dominant polycystic kidney disease. Importantly, previous studies have measured mainly intact FGF23 (iFGF23) rather than investigating the cleavage ratio involving C-terminal fragments (cFGF23).
Currently, there is no substantial evidence that CKD etiology meaningfully affects the iFGF23:cFGF23 ratio. Therefore, to maintain sufficient statistical power and clarity of analysis, and given the novelty of focusing on this ratio rather than absolute FGF23 levels, we decided to analyze CKD patients as a group without subdividing by etiology.
We acknowledge that future studies with larger cohorts and more detailed etiologic classification may help clarify potential etiologic influences on FGF23 processing. For now, our approach reflects the current gap in evidence and the primary goal of the study.
2. The goal of this study is to distinguish the difference between AKI and CKD, I feel confused about why the authors put the 64 patients with AKI and the 24 patients with acute on chronic kidney disease together in the same study group?
In our study, patients with chronic kidney disease (CKD) who experienced an abrupt worsening of kidney function, meeting KDIGO criteria for acute kidney injury (AKI) and having an identifiable precipitating factor (e.g., sepsis, hypovolemia), were classified as AKI-on-CKD. This subgroup was analyzed together with patients with de novo AKI because both groups represent acute kidney injury episodes that share similar pathophysiology, clinical implications, and require analogous diagnostic and therapeutic approaches.
We have now clarified this rationale explicitly in the manuscript (see: Methods, section: Study design and participants), hoping it addresses the concern and justifies the combined analysis.
3. In general, the performance of a diagnostic maker is evaluated using the Receiver Operating Characteristic (ROC) curve. The author should add a curve analysis.
As recommended, we have performed a ROC curve analysis to evaluate the diagnostic performance of the marker. The results, including the ROC curve, AUC, and optimal cut points with corresponding sensitivity and specificity, have now been added to the revised manuscript (see Results, Figure 4). We believe this addition strengthens the diagnostic evaluation of our biomarker as suggested.
4. The author says “adjusted analyses do not support its utility as a robust diagnostic marker for distinguishing AKI from CKD” I think the conclusion diminish academic and innovative value.
We acknowledge that our statement may appear strong; however, it accurately reflects the actual results of our analyses, which did not demonstrate sufficient diagnostic performance of FGF23 to reliably differentiate AKI from CKD after adjusting for relevant confounders. We intended to provide an accurate interpretation of the findings. Nevertheless, our work underscores a critical message for the field: the evaluation of novel biomarkers must involve sophisticated statistical methods that can dissect the complex interplay of confounding, clinically significant factors present in ill patients. We added this statement in the Discussion section.
Minor issues:
1. Page 4, Lines 158-160: “From this cohort, 88 patients (50,9%) were classified into the study group, which included 64 individuals with acute kidney injury (36.99% of the study group) and 24 individuals with significant exacerbation of chronic kidney disease (13.87%).” The author wrote “percentage of the study group,” but it should be “percentage of the whole group.” The proportion of the 64 acute kidney injury patients with the study group should be 72.7%.
Thank you for your comment. We have resigned from using the terms 'study group' and 'control group.' Instead, we refer to these as the 'AKI group' and the 'CKD group' throughout the manuscript. Additionally, in the specified section, we have replaced the term 'study group' with 'study population' to improve clarity and consistency.
2. Page 4, Line 157: “enrolled on the study” should be changed to “enrolled in the study”.
Thank you for your vigilance. We corrected the mistake.
3. Page 4, Line 158: “50,9%” should be changed to “50.9%”.
Thank you for your vigilance. We corrected the typo.
4. The sample sizes for the AKI group (n=88) and the CKD group (n=85) are too small. It can decrease statistical power.
We fully agree that the relatively small sample sizes of both groups may limit the statistical power and generalizability of our findings. As stated in the Discussion section, this limitation was acknowledged in the manuscript. To address your concern more explicitly, we have revised the text to further emphasize this point and to underscore the need for larger, multicenter studies to confirm our results.
Reviewer 3 Report
Comments and Suggestions for Authors
Although the iFGF23:cFGF23 ratio cannot serve as a reliable diagnostic marker for distinguishing AKI from CKD, this manuscript offers novel comparative insights into FGF23 metabolism across these conditions. A few suggestions for the authors:
- Clarify the criteria used to define “significant exacerbation of CKD” and explain why these patients were grouped together with de novo AKI cases.
- Include a flow diagram detailing patient inclusion and exclusion.
- Consider revising the manuscript title to reflect that the ratio itself lacks diagnostic reliability.
- Although discussed in the limitations that each ELISA measurement was performed only once for each sample, I still recommend including at least replicates or providing some tests to report the intra- and inter-assay CVs to demonstrate assay precision.
- In figures and figure legends, add sample sizes and significance markers.
- Correct the typo in line 328 (“enzyme-linked” instead of “enzym-linked”).
Author Response
Reviewer#3
Although the iFGF23:cFGF23 ratio cannot serve as a reliable diagnostic marker for distinguishing AKI from CKD, this manuscript offers novel comparative insights into FGF23 metabolism across these conditions. A few suggestions for the authors:
1. Clarify the criteria used to define “significant exacerbation of CKD” and explain why these patients were grouped together with de novo AKI cases.
We clarified the criteria for “significant exacerbation of CKD” by defining these patients as having chronic kidney disease with a sudden decline in renal function meeting KDIGO AKI criteria and an identifiable precipitating cause. We also explained in the Methods section that these cases were categorized as “AKI-on-CKD” and analyzed together with de novo AKI due to their standard pathophysiological features and clinical implications.
2.Include a flow diagram detailing patient inclusion and exclusion.
Thank you for your suggestion. We have added a flow diagram to illustrate the patient inclusion process (see revised Figure 4). All included patients had complete clinical and biochemical data and provided written informed consent. However, information regarding the total number of patients screened but not enrolled was not systematically collected, which we acknowledge as a limitation of the study and have stated in the Methods section.
3. Consider revising the manuscript title to reflect that the ratio itself lacks diagnostic reliability.
Thank you very much for your suggestion regarding the manuscript title. We agree that the diagnostic value of the iFGF23:cFGF23 ratio is limited based on our results, and it is crucial that the title accurately reflects this. Accordingly, we have revised the manuscript title to more precisely indicate that the ratio lacks sufficient diagnostic reliability, as suggested. The new title is: iFGF23:cFGF23 ratio is a questionable diagnostic marker for distinguishing between acute and chronic kidney disease.
4. Although discussed in the limitations that each ELISA measurement was performed only once for each sample, I still recommend including at least replicates or providing some tests to report the intra- and inter-assay CVs to demonstrate assay precision.
We fully agree that replicate measurements and reporting intra- and inter-assay coefficients of variation (CVs) would further strengthen the reliability of the results. Unfortunately, due to the limited volume of biological material and funding constraints, we were not able to perform replicate measurements. As mentioned in the Discussion section, each ELISA measurement was performed once per sample. However, all analyses were performed using commercially available, validated ELISA kits strictly following the manufacturer’s instructions, which include quality controls to ensure assay accuracy. We have emphasized this point in the revised manuscript.
5. In figures and figure legends, add sample sizes and significance markers.
Thank you very much for this valuable suggestion. We have revised all figures and figure legends to include information on sample sizes and appropriate significance markers, as recommended. We believe these additions increase the clarity and transparency of the presented results.
6. Correct the typo in line 328 (“enzyme-linked” instead of “enzym-linked”).
Thank you for your vigilance. We corrected the typo.
Round 2
Reviewer 1 Report
Comments and Suggestions for Authors
I
One more issue needs to be addressed for proper interpretation of the results.
Although there is no available data on iron status, it is important to know whether any of the included subjects received iron infusions, as these may be associated with increased FGF-23 levels (https://doi.org/10.1080/0886022X.2022.2164305) and could potentially influence the findings of this study.
Please either provide data regarding this issue or acknowledge it as a limitation
Author Response
Comment: One more issue needs to be addressed for proper interpretation of the results. Although there is no available data on iron status, it is important to know whether any of the included subjects received iron infusions, as these may be associated with increased FGF-23 levels (https://doi.org/10.1080/0886022X.2022.2164305) and could potentially influence the findings of this study. Please either provide data regarding this issue or acknowledge it as a limitation
Answer: Thank you for raising this important point. None of the patients included in this study received intravenous iron supplementation prior to blood sampling. This information has been added to the Methods section for clarity.
Reviewer 2 Report
Comments and Suggestions for Authors
Thank you for the thorough revisions. The authors have successfully addressed the major and minor points from the first round of review, particularly with the addition of the ROC analysis and the clarifications to the methodology. The manuscript is now suitable for publication.
Author Response
Comment: Thank you for the thorough revisions. The authors have successfully addressed the major and minor points from the first round of review, particularly with the addition of the ROC analysis and the clarifications to the methodology. The manuscript is now suitable for publication.
Response: We sincerely thank you for the positive evaluation and constructive comments provided during the review process. The feedback has greatly contributed to improving the quality and clarity of our manuscript.
Reviewer 3 Report
Comments and Suggestions for Authors
Most of my concerns have been addressed. However, although authors mentioned Figure 4 in the manuscript and the response, I did not see it in the manuscript.
Author Response
Comment: Most of my concerns have been addressed. However, although authors mentioned Figure 4 in the manuscript and the response, I did not see it in the manuscript.
Response: Figure 4 was provided in a separate attachment during the previous submission but was inadvertently not embedded within the main manuscript file. This has now been corrected, and Figure 4 is included in the revised manuscript in the appropriate section.